# WorldQA: Multimodal World Knowledge in Videos through Long-Chain Reasoning

## Abstract

Multimodal information, together with our knowledge, help us to understand the complex and dynamic world. Large language models (LLM) and large multimodal models (LMM), however, still struggle to emulate this capability. In this paper, we present **WorldQA**, a video understanding dataset designed to push the boundaries of multimodal world models with three appealing properties: **(1) Multimodal Inputs:** The dataset comprises 1007 question-answer pairs and 303 videos, necessitating the analysis of both auditory and visual data for successful interpretation. **(2) World Knowledge:** We identify five essential types of world knowledge for question formulation. This approach challenges models to extend their capabilities beyond mere perception. **(3) Long-Chain Reasoning:** Our dataset introduces an average reasoning step of 4.45, notably surpassing other videoQA datasets. Furthermore, we introduce **WorldRetriever**, an agent designed to synthesize expert knowledge into a coherent reasoning chain, thereby facilitating accurate responses to WorldQA queries. Extensive evaluations of 13 prominent LLMs and LMMs reveal that WorldRetriever, although being the most effective model, achieved only 70% of human-level performance in multiple-choice questions. This finding highlights the necessity for further advancement in the reasoning and comprehension abilities of models. Our experiments also yield several key insights. For instance, while humans tend to perform better with increased frames, current LMMs, including WorldRetriever, show diminished performance under similar conditions. We hope that WorldQA, our methodology, and these insights could contribute to the future development of multimodal world models.

## 1 Introduction

Consider the scene in Fig. 1. It shows more than a woman simply drinking coffee and picking up clothes. With the background sounds of a ticking clock and a mix of radio broadcasts, along with the noise of a door opening and closing, we naturally form a story: she's just waken up and is getting ready, probably for work. Understanding this video requires combining two key human skills: perception and cognition. Perception lets us notice and recognize details, like the clock's time, and the radio's sound. Cognition, on the other hand, involves using knowledge from our own experiences, like knowing typical work hours. Together, these skills enable us to follow the video's story through a logical series of steps.

For humans, merging perception and cognition to understand video narratives is intuitive, but what about Large Multi-modal Models (LMMs)? To push the boundaries of comprehensive video understanding, we introduce **WorldQA**, a diagnostic benchmark dataset challenges machines to answer questions about a video by employing multimodal data and world knowledge. WorldQA is distinguished by three main features: **(1) Multimodal Video Input:** Success requires analyzing both auditory and visual data. **(2) Emphasis on World Knowledge:** Questions in the dataset necessitate engagement with broad world knowledge. We identify five knowledge types critical for question answering: societal norms, multimodal associations, self-motivation, tool use, and social interactions, detailed in Sec. 3. **(3) Long-Chain Reasoning:** The dataset promotes integrating multimodal information and world knowledge across frames for complex reasoning. Currently, the dataset includes 1007 question-answer pairs and 303 videos, with more details in Sec. 3. An initial analysis using GPT-4 OpenAI (2023) shows the average reasoning steps in WorldQA to be 4.45, notably

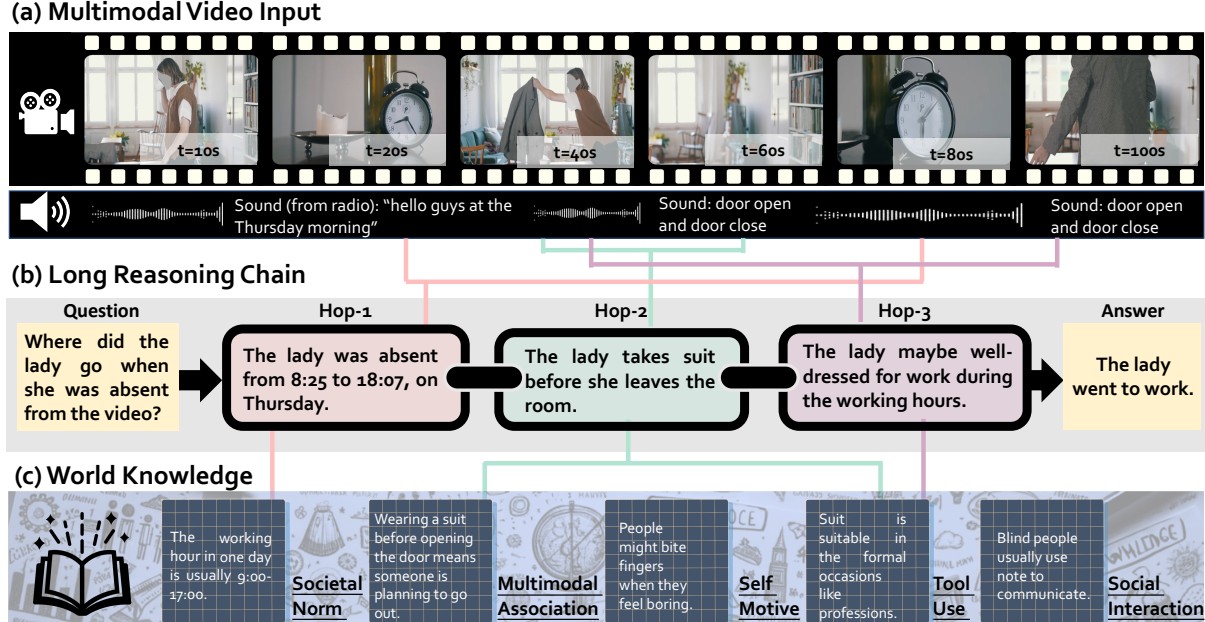

Figure 1: **An example video from our WorldQA.** To determine *where the lady went when she was absent from the video*, we rely on visual cues, auditory hints, and the application of world knowledge. This forms a reasoning chain to deduce the answer. WorldQA comprises 1007 question-answer pairs and 303 videos, spanning five types of world knowledge. On average, the reasoning chain consists of 4.45 steps.We recommend watch the video: https://www.youtube.com/watch?v=NXbJLLf9E_I

higher than the under-two average in other datasets, as demonstrated in Table 5. Evaluation protocols are discussed in the *Appendix* 3.1.

We propose exploring WorldQA with **WorldRetriver**, employing large language model (LLM) agents Shen et al. (2023); Surís et al. (2023); Chen et al. (2023); Wu et al. (2023). WorldRetriver breaks down each question into perception- or cognition-oriented tasks. These tasks are then addressed by specialized models—a **multimodal key information retriever** and a **world knowledge retriever**. The LLM integrates their outputs to form a cohesive reasoning chain, answering the question.

Our study presents a comprehensive evaluation of WorldQA, benchmarking WorldRetriver against 13 leading large language models (LLMs) and large multimodal models (LMMs), as well as comparing it to human performance. We focus on two tasks: open-ended and multiple-choice QA. WorldRetriver demonstrates superior performance in both areas, achieving 36.38% accuracy in open-ended QA and 36.59% in multiple-choice QA, surpassing even GPT-4V Yang et al. (2023b). Key findings include: (1) While WorldRetriver generally outperforms GPT-4V, the latter excels in questions that require complex reasoning, particularly at reasoning steps 8, 9, and 10. (2) Current open-source LMMs exhibit challenges with "consistency" in multiple-choice QA, as discussed in Sec.4.3. (3) Employing GPT-4 to evaluate open-ended QA models correlates well with human judgments, explored in Sec.4.5. (4) Although human performance typically improves with additional frames, our WorldQA and current open-source LMMs show decreased performance under similar conditions, detailed in Sec. 4.5.

## 2 Related Work

### 2.1 VideoQA Datasets

Visual Question-Answering (QA) Antol et al. (2015) is a key task in video-language research, spanning a wide range of datasets. These datasets evaluate various capabilities from basic visual perception, including

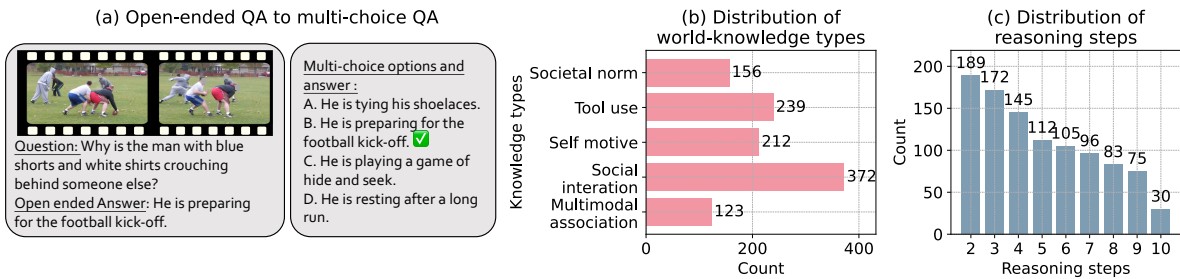

Figure 2: (a) An example for reformating open-ended QA into multi-choice QA. (b) the distribution of different world-knowledge types. (c) the distribution of reasoning step counts.

Table 1: **Dataset comparisons.** Reason. stands for reasoning. Avg. stands for average. Q/A stands for question and answer.

| Dataset | Multi modal? | World knowledge? | Avg. reason. steps | Avg. length (Q/A) |
|---|---|---|---|---|
| MSVD-QA | ✗ | ✗ | 1.40 | 6.6/1.0 |
| TGIF-QA | ✗ | ✗ | 1.71 | 8.7/2.1 |
| TVQA | ✗ | ✗ | 1.91 | 13.4/5.3 |
| ActivityNet-QA | ✗ | ✗ | 1.62 | 8.7/1.2 |
| NExT-QA | ✗ | ✗ | 1.31 | 11.6/2.9 |
| Social-IQ | ✓ | ✗ | 1.98 | 11.7/11.4 |
| WorldQA | ✓ | ✓ | **4.45** | **14.2/24.3** |

activity recognition Yu et al. (2019), concept detection Xu et al. (2017), and counting Jang et al. (2017), to more advanced visual reasoning such as compositional Grunde-McLaughlin et al. (2021), causal Xiao et al. (2021); Yi et al. (2019); Xu et al. (2021), and situated reasoning Wu et al. (2021). Beyond using visual information for answering questions, KnowIT Garcia et al. (2020) incorporates external knowledge from the "Big Bang Theory" in its question design, whereas Social-IQ Zadeh et al. (2019) leverages both visual and auditory modalities, solving human-centric questions, Unlike KnowIT, which is limited to knowledge from a single TV series, WorldQA draws on a broader range of general world knowledge. Additionally, WorldQA expands beyond the human-centric focus of Social-IQ to encompass a variety of subjects including animals and machines.

Moreover, our analysis of existing videoQA datasets identifies a notable limitation: most require less than two reasoning steps per question, highlighting a gap in their ability to facilitate complex reasoning. To address this, we introduce WorldQA, a dataset specifically designed to challenge models with more intricate reasoning sequences.

## 2.2 Vision-Language Models

Recent developments in large language models (LLMs) such as GPTs Radford et al. (2019); Brown et al. (2020), LLaMA Touvron et al. (2023), and Vicuna Chiang et al. (2023) have enhanced the efficacy of vision-language models like Flamingo Alayrac et al. (2022); Awadalla et al. (2023) and FrozenBiLM Yang et al. (2022), especially in zero-shot learning contexts. Researchers are now exploring instruction-tuned models, including Otter Li et al. (2023a), InstructBLIP Dai et al. (2023), , LLaVA Liu et al. (2023b) and others Ye et al. (2023); Li et al. (2023b); Zhang et al. (2023); Muhammad Maaz & Khan (2023); Gao et al. (2023), to enhance the interaction abilities of these vision-language models. These models excel in intricate human-model interactions and are ideal for multi-modal chatbot applications.

**System Message**

As an AI visual assistant, your objective is to determine the number of reasoning steps—each representing a logical or causal link—between a given question and paired answer. In this context, a reasoning step elucidates the process through which the answer relates to the question. The goal is to quantify the depth of reasoning, whereby a direct causation between the answer and question constitutes one step, while the presence of intermediate steps to establish a connection warrants a higher count.

Let's begin this task, you reasoning steps should be as fewer as possible. In your reasoning step, do not generate anything unmentioned in the question and answer.

You format should be :

Reasoning steps: <reasoning steps>

Number of Reasoning steps:

<number of Reasoning steps>

Question: {question}

Answer: {answer}

---

**In-context Examples**

Question: why did the kid drink water?

Answer: the kid is thirsty

Reasoning steps:

1. When people feel thirsty, they want to drink water

Number of Reasoning steps:1

Question: Why did the kid touch the cup? Answer: the kid is thirsty

Reasoning steps:

1. When people feel thirsty, they want to drink water.

2. Cups are usually used to collect water.

Number of Reasoning steps:2

Question: Why does the tank turn red?

Answer: It stands to reason that there was a piranha in the tank, The piranha bit the man in the tank. The person in the water tank bleeds because of this. Blood turns the water tank red.

Reasoning steps:

1. There was a piranha in the tank.

2. Piranha bit the man in the tank.

3. The person in the water tank bleeds because of this.

4. Blood turns the water tank red.

Number of Reasoning steps:4

Table 2: System message and in-context exemplars for reasoning step prompting.

## 3 WorldQA Dataset

In this section, we detail the WorldQA, aimed at creating a benchmark in video question answering for evaluating artificial intelligence (AI) in complex reasoning. The WorldQA comprises 303 videos and 1007 questions. As WorldQA is used solely for evaluation purposes, we believe the number of questions is sufficient. We first describe our thorough annotation process and multiple validation stages, then provide detailed statistics of the WorldQA.

### 3.1 Prompt for Reasoning Hop

The prompt for getting the required reasoning hop for each question is shown in Table. 2.

### 3.2 Prompt for Multi-choice QA

The prompt for getting the required reasoning hop for each question is shown in Table. 3.

### 3.3 Prompt for Answer Composition

**Video Acquisition Stage**   The WorldQA sources data from two components: (1) PVSGYang et al. (2023a), which comprises 268 third-person videos from the VidOR dataset Shang et al. (2019) along with 200 egocentric videos sourced equally from Epic-Kitchens Damen et al. (2018) and Ego4D Grauman et al. (2022), and (2)

**System Message**
As an AI visual assistant, your task involves first analysing video content, including various textual descriptions, and then rewriting the given question into multi-choice questions (with four options and only correct answer) related to the video.
##Video Textual Description##
1. Event Descriptions: Descriptions of specific video segments identified by their timestamps (start and end times).
##Guidelines##
While executing this task, please adhere to the following guidelines:
1. The answer to each question should be in the form of a single letter: A, B, C, or D.
2. All the options you provide should be roughly the same length.
3. The choices you present should be formulated in a way that makes them tricky to differentiate, thus creating some confusion for the individual answering the question.
4. You should rewriting the given question even if you think it do not seem to match the content described.
5. The correct option should follow the answer of the given question.
6. Starting your response without saying anything unrelated to the output format.
##Format## Your output format should be like:
Question:
<question>
Option
A.XXX
B.XXX
C.XXX
D.XXX
Answer:
<answer>
Let's begin this task, you should rewrite all the below questions (in the (2)) into multi-choice question, based on these video event descriptions(in the (1)).
(1) Event Descriptions:
{event_descriptions}
(2) The questions for rewriting:
Question: {question}
Answer: {answer}

Table 3: System message for creating multi-choice question. {XXX} is the placeholder for specific information, *i.e.,* , event description, question and answer.

**System Message**
As an AI visual assistant, your task involves synthesizing an answer to a question about a video by integrating responses from two expert models: Model A and Model B. Model A provides the initial answer to the question, while Model B focuses on contributing additional external knowledge to solve the question.
Let's begin this task:
Question:
{question}
Response from Model A:
{Model_A_response}
Response from Model B:
{Model_B_response}
Step-by-step reasoning:
<reasoning process>
Answer:
<answer>
Let's think step by step.

Table 4: System message for creating answer composition.{XXX} is the placeholder for specific information, *i.e.,* , question, response from Model A and Model B.

user-generated content from YouTube, identified using search terms such as "1 minute movie" and "short movie". In total, this initial dataset encompasses 1000 videos.

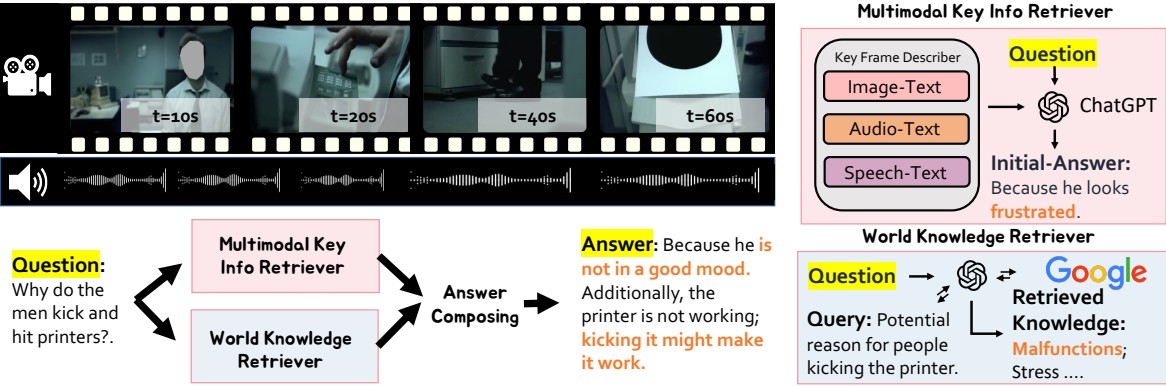

Figure 3: **WorldRetriever**, an agent designed to synthesize expert knowledge into a coherent reasoning chain for answering questions.

**Question and Answer Creation Stage**   Expert annotators formulated questions to test two aspects of QA systems:

**1) World Knowledge Understanding**: Understanding video content goes beyond the perception level, requiring an integration of broad world knowledge. This includes: *a) Tool Use*: Understanding the purposes of various tools and concepts, *e.g.,* , recognizing that a hammer is for driving nails. *b) Societal Norms*: Comprehending behaviors, traditions, and unwritten rules within societies, *e.g.,* , the custom of handshaking in certain cultures. *c) Self Motive*: Identifying personal intentions and motivations, *e.g.,* , eating to satisfy hunger. *d) Social Interaction*: Understanding the subtleties of communication and relationships, such as interpreting social signals. *e.g.,* , recognizing that people who cannot speak may use written notes to communicate. *e) Multimodal Association*: Linking vision and hearing to form a complete understanding. *e.g.,* , sound of alarm bells with the visual of people evacuating to infer a fire.

**2) Long Reasoning Chain**: Deducing "Who is the murderer?" in a detective movie involves complex reasoning steps. However, current videoQA, like NExT-QA Xiao et al. (2021) and Social-IQ Zadeh et al. (2019), often feature basic questions that require minimal reasoning. For example, typical NExT-QA questions, such as "Why are the dogs running around?" with an answer like "Chasing each other", require very few reasoning steps. Our analysis using GPT-4 found the average reasoning depth in NExT-QA to be 1.31, validating this observation.[1] WorldQA, aims to challenge this by including questions that demand multi-step reasoning, where at least two logical steps are necessary to arrive at the answer.

It's important to note that a single question might involve multiple types of world knowledge. Annotators were tasked with providing answers to these questions. To broaden the utility of our dataset, we used GPT-4 OpenAI (2023) to transform our question-answer pairs into a multiple-choice format, as shown in Fig. 2(a).[2]

**Question and Answer Validation**   To refine our question set, we established criteria for deletion as follows: (1) Questions that do not require world knowledge, as verified by an annotator different from the question creator; (2) Questions for which GPT-4/ChatGPT responses align with human-annotated answers, as detailed in Section 4.2; and (3) Questions that are solvable in fewer than two reasoning steps, initially verified by an annotator different from the question creator and then further filtered by querying GPT-4, as described in 3.1. This process yielded a dataset of 303 videos and 1007 questions.

In improving multi-choice questions, we aimed to: (1) make all options similar in length, and (2) ensure questions can only be correctly answered by models with visual input. To achieve the second goal, we repeatedly adjusted each option until GPT-4/ChatGPT could not answer correctly. As Table 5 shows, all

---

[1]The details of the prompt are shown in the 3.1.

[2]The details of the prompt are shown in the 3.2.

LLMs (Language Learning Models) scored zero on these questions. However, GPT-4 achieved 35.34 points in the NExT-QA multi-choice test, as explained in the 4.5.

### 3.4 Data Statistics

**Dataset Comparison**  WorldQA presents significant advancements over existing datasets, as Table 1 illustrates. Firstly, it requires complex multi-step reasoning. Using GPT-4, we evaluated the reasoning steps in each question-answer pair across datasets; WorldQA averages 4.45 steps, notably higher than others which typically involve less than two steps. This complexity is also evident in answer lengths: while answers in other VideoQA datasets average below five words, those in WorldQA average 24.3 words. Secondly, it necessitates more than visual information for success. WorldQA encompasses audio comprehension and world knowledge, expanding its scope beyond mere video visuals for effective question resolution. To our knowledge, it represents the first VideoQA dataset that incorporates questions necessitating world knowledge.

**Knowledge Types and Reasoning Step Statistics**  As illustrated in Fig. 2(b), the majority of these questions fall under the "social interaction" category. Furthermore, Fig. 2(c) demonstrates that the reasoning steps in WorldQA vary, ranging from two to ten steps.

## 4 Experiments

### 4.1 Experimental Setup

Our study assesses the performance of diverse models on WorldQA across four settings: **(1) Large Multimodal Models (LMMs) for Video Processing:** This category includes FrozenBiLM Yang et al. (2022), Otter-VideoLi et al. (2023a), VideoChatLi et al. (2023b), Video-LLaMAZhang et al. (2023), Video-ChatGPTMuhammad Maaz & Khan (2023), and mPLUG-OwlYe et al. (2023). These open-sourced models, which are trained with a specific number of frames, typically combine a language model (*e.g.,* , LLamaTouvron et al. (2023) or VicunaChiang et al. (2023)), a vision encoder (*e.g.,* , CLIPRadford et al. (2021)), and a connector (*e.g.,* , linear layer Liu et al. (2023a)) to convert vision embeddings into "text tokens." Additionally, we include the recently API-accessible GPT-4V in our analysis. **(2) LMMs for Image Inputs:** This group comprises Qwen-VLBai et al. (2023) and LLaVA-1.5Liu et al. (2023a). **(3) Large Language Models (LLMs):** We evaluate models such as Vicuna-v1.5-7BChiang et al. (2023) (abbreviated as Vicuna), ChatGPTOpenAI (2023), and GPT-4OpenAI (2023), focusing on scenarios where their inputs consist solely of the question or the question accompanied by options. **(4) WorldRetriver:** This approach uses ChatGPT as its predefined LLM and LLaVA-v1.5-7BLiu et al. (2023a) (abbreviated as LLaVA) for image-text tasks, BeatsChen et al. (2022) for audio-text, and WhisperRadford et al. (2023) for speech-to-text conversion. We use LLaVA to describe images, selecting eight frames uniformly. Audio clips are extracted from videos using PydubRobert et al. (2018) and analyzed with Beats. Our approach also integrates ReACTYao et al. (2022) for world knowledge retrieval. **(5) Augmented LLM and Human Performance:** Inspired by MathVista Lu et al. (2023), our study evaluates human performance alongside three augmented LLMs: Augmented Vicuna/ChatGPT/GPT-4. As mentioned above, the current LMM consists of a language model, a vision encoder, and a connector. We propose that LMM performance might be limited by the vision encoder and connector's ability to translate visual data into "text tokens." To explore this hypothesis, we converted video information into event descriptions annotated by humans (for detailed information on event descriptions, we prompted the language models with these descriptions alongside questions to get responses, in what we term "augmented LLM." This experiment helps us estimate the potential maximum performance for LMMs using similar language models.

Notably, except for GPT-4V and FrozenBiLM, the other LMMs use a 7B language decoder, similar in size to Vicuna. Video-LLaMA uniquely processes both audio and visual modalities.

### 4.2 Open-Ended QA

**Definitions and Metrics**  Recent studies Xiao et al. (2021); Zheng et al. (2023) have increasingly turned their attention to generation-based open-ended QA, where answers are not confined to a closed set. However,

Table 5: **Evaluation of Large Multimodal Models (LMMs), Language Models (LLMs), and WorldRetriver in WorldQA.** We introduce two upperbounds for comparison: LLMs augmented with huamn-annotated event descriptions, labeled as (Aug.) X, and the human performance. The best model in each group is highlighted in blue, while the overall top performer in all tasks is marked in red. Different types of inputs include: $Q$ for Question, $V$ for Video, $I$ for Image, and $V_d$ for Video Description. Langauge model param. indicates the parameter of the language model.

| Model (language model param.) | Input | Open ended ↑ | Multi choice ↑ |
|---|---|---|---|
| *Large Multimodal Models (LMMs)-Video* | | | |
| FrozenBiLM (900M) | $Q, V$ | 8.21 | 0.32 |
| Otter-Video (7B) | $Q, V$ | 24.22 | 6.11 |
| VideoChat (7B) | $Q, V$ | 24.43 | 1.29 |
| LLaMA-Adapter (7B) | $Q, V$ | 25.87 | 12.04 |
| Video-LLaMA (7B) | $Q, V$ | 26.80 | 4.81 |
| Video-ChatGPT (7B) | $Q, V$ | 28.51 | 13.25 |
| mPLUG-Owl (7B) | $Q, V$ | 31.89 | 0.75 |
| GPT-4V(ision) (-) | $Q, V$ | 35.37 | 32.83 |
| *Large Multimodal Models (LMMs)-Image* | | | |
| Qwen-VL (7B) | $Q, I$ | 24.04 | 12.80 |
| LLaVA (7B) | $Q, I$ | 31.31 | 0.30 |
| *Large Language Models (LLMs)* | | | |
| Vicuna (7B) | $Q$ | 22.44 | 0.00 |
| ChatGPT (20B) | $Q$ | 24.24 | 0.00 |
| GPT-4 (-) | $Q$ | 28.73 | 0.00 |
| *LLM Agent* | | | |
| Ours (ChatGPT as LLM) (20B) | $Q, V$ | 36.38 | 36.59 |
| *Upper Bound with Human Transcription* | | | |
| (Aug.) Vicuna (7B) | $Q, V_d$ | 38.71 | 23.90 |
| (Aug.) ChatGPT (20B) | $Q, V_d$ | 46.50 | 46.06 |
| (Aug.) GPT-4 (-) | $Q, V_d$ | 48.46 | 56.06 |
| *Human-Level Performance* | | | |
| Human | $Q, V$ | 72.43 | 88.79 |

assessing the quality of open-ended text remains challenging. For instance, current evaluation protocols like NExT-QA may overlook semantic correlations; a response "a cute teddy bear" receives no credit if the ground truth is "a teddy bear." This issue is compounded as answer length increases.

In the field of natural language generation (NLG), recent initiatives have investigated the use of GPT-4 for assessing the quality of open-ended model-generated responses Zheng et al. (2023); Xie et al. (2023). For evaluating open-ended QA, we also use GPT-4. Our scoring system for model answer $A$ against ground truth $G$ is: (1) $A = G$: Correct (1 point), (2) $A \cap G = \emptyset$: Incorrect (0 points), (3) $\emptyset < A \cap G < A \cup G$: Partially correct (0.3 points), (4) $A \subset G$, $A \neq G$: Incomplete but correct (0.5 points), (5) $G \subset A$, $A \neq G$: Redundant (0.5 points). GPT-4's scoring is exemplified in Fig. 5.

**Main Results** Our open-ended QA evaluation revealed several key insights: **1) Superiority of WorldRetriver and GPT-4V:** Our method, WorldRetriver, using ChatGPT, surpasses the ChatGPT baseline by 12.14 points and GPT-4V by 1.01 points, as shown in Table. 5. In Fig. 4, WorldRetriver and GPT-4V outperform the leading open-source LMM, mPLUG-Owl, particularly in reasoning beyond five steps. While WorldRetriver beats GPT-4V in the overall performance, the latter shows strength in complex reasoning tasks (steps 6 to 9). However, the closed-source nature of GPT-4V suggests future research opportunities to

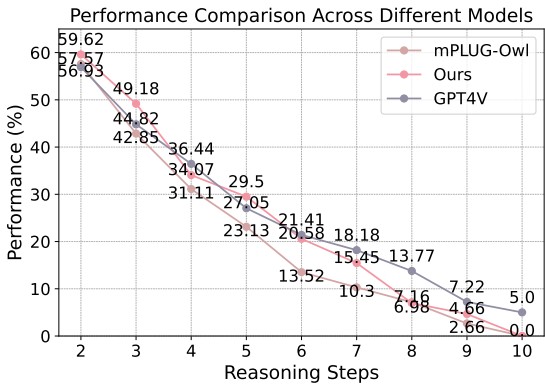

Figure 4: Comparative performance of advanced LMM and our method across increasing reasoning steps.

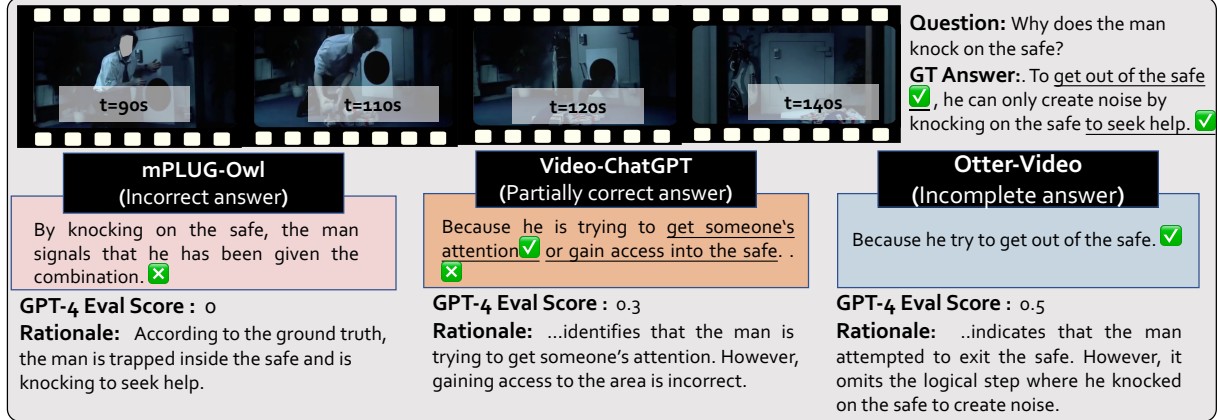

Figure 5: Examples of how does GPT-4 score in the open-ended question.

understand these differences. **2) Non-Zero Performance in Large Language Models (LLM):** Although WorldQA is a videoQA dataset, it is observable that LLMs can still achieve a certain level of accuracy by using only the questions as input, which we consider reasonable. For instance, in response to the question, "What did the lady do when she left home?", an LLM might reply, "She may have gone shopping or to work." While this response is not entirely accurate, it closely approximates the actual answer, "She went to work." However, as mentioned in Sec. 3, questions that could be answered with 100% accuracy by LLMs were excluded. **3) Limitations of Current Video-Based LMM in Handling Multiple Frames:** We found that the image-based LLaVA model outperforms most video-based models in performance, which is surprising given that video-based LLMs are capable of processing multiple frames. This leads to a pivotal question: Do the tasks in WorldQA require only a minimal number of frames, or do current video-based LMMs struggle to use multiple frames effectively? Sec. 4.5 presents experimental evidence supporting the latter. **4) Impact of Language Model Size:** When compared to LMMs with a 7B-language model, Vicuna's lower performance is expected due to its lack of vision information for answering questions. However, GPT-4 surpasses six out of the eight LMMs. This underscores the significant benefits of GPT-4's advanced language processing capabilities. It also suggests that incorporating a languge model as powerful as GPT-4's could significantly advance the capabilities of current LMMs. **5) Potential for Improvement:** By employing the same language model, namely Vicuna, our augmented Vicuna model surpasses LLaVA by 7.4 points. This result underscores a significant opportunity for improving LMMs. Moreover, even the augmented GPT-4 reaches merely 67% of human performance, suggesting a considerable scope for advancing current models.

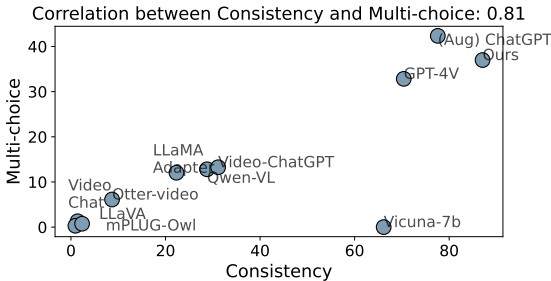

Figure 6: Analysis of the correlation between multi-choice QA performance of models and their consistency, defined as the frequency of selecting the same response $N$ times across varied sequences of $N$ options.

## 4.3 Multi-Choice QA

**Definitions and Metrics** Compared to open-ended QA, multi-choice QA tasks simplify the QA task, as the correct answer is always among the provided options. We follow the approach of MMBench Liu et al. (2023c) and use its proposed CircularEval evaluation method to evaluate model performance. CircularEval requires the model to answer each question $N$ times, where $N$ is the number of choices. Each iteration involves a circular shift of the options, creating a different arrangement. This technique mitigates the effect of random guessing, which could otherwise lead to a 25% Top-1 accuracy rate in scenarios with four choices. If the model's response does not match any of the given options (*e.g.,* , A, B, C, D), ChatGPT evaluates semantic similarity to determine the most appropriate choice. For more details on CircularEval, please refer to MMBench.

**Main Results** In Table 5, WorldRetriver performs better than other models in multi-choice QA, showing a notable 3.76-point lead over GPT-4V. Also, we made sure each multi-choice question could only be correctly answered by models that can process visual input, resulting in LLMs scoring zero. Our analysis highlights important insights: **1) Consistency Challenge in Open-Source LMMs:** Notably, while mPLUG-Owl and LLaVA show strong performance in open-ended tasks, their effectiveness decreases in multi-choice QA scenarios. We propose that this decline is likely due to inconsistent choice when the order of options is varied. To illustrate this issue, we introduce the "consistency rate" metric in Fig. 6, which quantifies how often a model selects the same option across different arrangements of the $N$ options, regardless of the answer's correctness. As demonstrated in this figure, there is a significant correlation between the consistency rate and accuracy in multi-choice QA. This finding highlights the importance of a model's ability to consistently select the same answer as a key indicator of its proficiency in CircularEval. **2) Consistency Loss in Fine-Tuned Language Model:** Among LMMs, LLaVA and mPLUG-Owl are unique in tuning their language models during the instruction tuning phase. However, this approach results in notably poorer consistency and, consequently, inferior performance in multi-choice QA tasks. A direct comparison between LLaVA, which uses the Vicuna, and Vicuna itself reveals a significant drop in consistency for LLaVA. This suggests that tuning language decoders during the instruction tuning stage can adversely affect the model's overall consistency. **3) Potential for Improvement:** With human benchmarks at 88.79, there is substantial scope for models to match or exceed human performance in video comprehension.

## 4.4 Ablation Studies of WorldRetriver

In this subsection, we examine the impact of distinct components within the WorldRetriver.

|  | Open-ended ↑ | Multi-choice ↑ |
|---|---|---|
| Ours | 36.38 | 36.59 |
| - world-knowledge | 34.78 (-1.60) | 35.46 (-1.13) |
| - speech-text | 33.45 (-1.33) | 34.12 (-1.34) |
| - audio-text | 33.23 (-0.22) | 34.45 (+0.23) |
| - image-text | 30.98 (-2.25) | 2.23 (-32.22) |

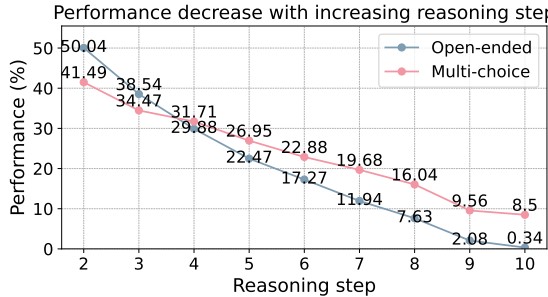

(a) Illustration of how performance declines as the number of reasoning steps increases, in both open-ended and multiple-choice tasks.

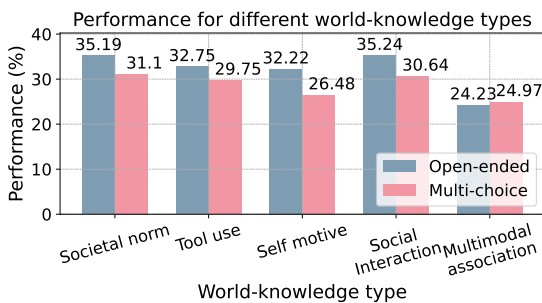

(b) A breakdown of performance across various knowledge types, highlighting the notably lower scores in the multimodal association category.

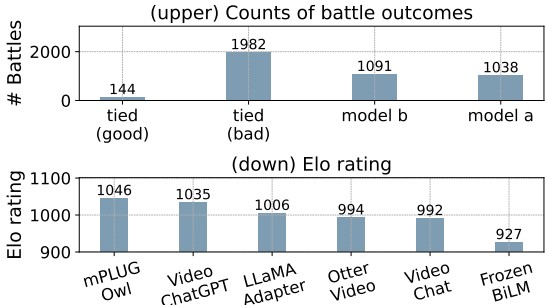

(c) (upper) Comparative counts of model choices from Human evaluators. (down) Elo scores assigned to various models from Human evaluators.

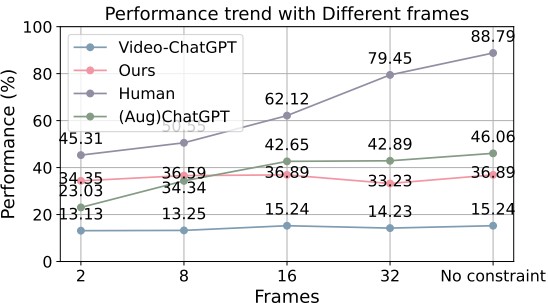

(d) Comparative analysis of model performance in multiple-choice QA tasks under varying frame constraints. "no constraint" for models indicates their optimal performance, whereas for humans, it denotes the ability to answer questions after viewing the entire video.

Figure 7: Key findings emerged from the further analysis.

As shown above, our findings indicate that the image-text model is the most critical, while the audio-text model contributes the least to overall performance. Notably, the enhancement provided by the audio-text component is negligible. To investigate the cause, we scrutinized the output of the audio-text model, which is responsible for categorizing audio within every clips in videos. The analysis reveals that the prevalent audio-text model, Beats, seldom produces accurate classification labels for video audio content. Additionally, leveraging the capabilities of expert models—Whisper and ReACT—might be the core reason why WorldRetriver outperforms GPT-4V.

## 4.5 Further Analysis

**The Relation of Reasoning Steps/Knowledge Type and Performance**   We investigated the impact of the number of reasoning steps on a model's ability to answer questions, specifically examining the correlation between reasoning steps and performance. In Fig. 7a, the results shown for each step are averaged from the models in LMMs-Video, LMMs-Image, WorldRetriver, and Augmented-LLM in Table 5. Our findings reveal that as the number of reasoning steps increases, performance deteriorates. Notably, by the time it reaches 10 reasoning steps, the average score is almost zero in open-ended tasks.

Additionally, we examined the average performance of models across five distinct word-knowledge types. As shown in Fig. 7b, it's notable that the poorest average performance is observed in the "multimodal association" word-knowledge. This is likely because current models are unable to handle audio/speech signals

**Does GPT-4's scoring align with human preferences?** In Sec. 4.2, a question arises: does the model that achieves the highest score in open-ended QA, as evaluated by the predefined GPT-4 scoring system, truly align with human preferences? To investigate this, we employed the Model Arena approach Zheng et al. (2023); Xu et al. (2023), which involves rating models based on human preferences in a side-by-side comparison of two model-generated answers. The Elo rating system, employed in Model Arena, aggregates these judgments to rank models. If the rankings in Table. 5 for the open-ended QA task correspond with those derived from the Model Arena, it would validate the effectiveness of the GPT-4 scoring system in evaluating LLM-generated responses.

In our study, we conducted five rounds of Model Arena for each question pertaining to six video-based LLMs, huamn judges reviewed related videos for making the decision. This procedure was replicated across 4,255 comparisons.

The results, shown in Fig. 7c, primarily classify the responses as "tied (bad)", suggesting that current LMMs fail to produce high-quality responses. A comparison of the Elo rankings with the open-ended QA performance in the LLMs-video section (Table 5) shows a significant correlation. This agreement with open-ended QA scores validates our evaluation method for open-ended answers.

**Do More Frames Impair Performance?** In Sec. 4.3, we pose a question: do the tasks in WorldQA require only a minimal number of frames, or do current LMMs struggle with effectively using multiple frames? Our experiments suggest the latter. As depicted in Fig. 7d, a distinct pattern is observed: the performance of humans and augmented ChatGPT—which selects event descriptions based on the time period of sampled frames—enhances on WorldQA when more frames are used. In contrast, Video-ChatGPT and our proposed methods exhibit peak performance at approximately 16 frames. This suggests a limitation in how current models process multiple frames.

**Can Uni-modal model answer?**

|  | Social-IQ | NextQA | KnowIT | Ours(WorldQA) |
|---|---|---|---|---|
| GPT-4 | 29.47 | 35.34 | 36.44 | 0.00 |

In Sec. 3.2, we mention that for multi-choice questions, we repeatedly adjusted each option until GPT-4/ChatGPT was unable to provide an answer. This approach ensures that each question can only be answered by models equipped with visual input capabilities. In contrast, we noted that many existing videoQA datasets contain questions that GPT-4 can answer correctly without visual information. We explored this issue using CircularEval, and the results of these experiments are detailed above.

## 5 Discussion and Conclusion

In this study, we introduce WorldQA, an innovative dataset designed to assess the ability of visual-language models in understanding videos. WorldQA distinctively emphasizes the integration of multimodal information and the application of world knowledge for complex reasoning, reflecting a crucial aspect of human intelligence. Concurrently, we present WorldRetriver, a technique inspired by human approaches to video understanding. Our experiments with WorldQA reveal that current models still fall short of human-like proficiency in video understanding.

## 6 Limitations

The videos in WorldQA are sourced from various platforms, including Ego4DGrauman et al. (2022), Epic-KitchensDamen et al. (2018), VidOR Shang et al. (2019), and YouTube. This diversity introduces potential biases inherent in these sources. Furthermore, there is a concern regarding the potential skew in the question-answer pairs, possibly influenced by the annotators' perspectives. Additionally, the significant observation that WorldRetriver struggles to process multiple frames highlights a crucial area for future improvement.

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
