# OpenReview forum: "WorldQA: Multimodal World Knowledge in Videos through Long-Chain Reasoning"
_TMLR — Rejected by TMLR_

### Review · Reviewer_DHbK · 2026-03-24

**Summary Of Contributions:**

Summary:

This paper presents WorldQA, a video understanding dataset designed to evaluate the reasoning ability of multimodal world models. The dataset contains multimodal inputs, and its questions are designed to require world knowledge and long-chain reasoning to answer correctly. The paper also introduces WorldRetriever, an agent that synthesizes world knowledge into coherent reasoning chains to provide accurate responses to WorldQA queries.

Strengths:

1. The paper clearly explains the motivation for building this dataset.
2. The paper includes several ablation studies and analyses that provide potentially interesting insights.

Weaknesses:

1. The paper’s structure could be better organized. Paragraphs, results, and figures are scattered, which makes the presentation harder to follow. For example, WorldRetriever is introduced on page 2 and then discussed again in Section 4.1 on page 7, while the figure illustrating its internal details and pipeline appears on page 6 without being clearly referenced in the surrounding text.
2. The role of world knowledge in WorldQA is not fully convincing. For instance, I am not convinced that the example in Figure 1 truly requires “world knowledge.” It seems closer to basic commonsense knowledge than to expert knowledge that would require internet access or database retrieval. It is therefore possible that the observed gains attributed to “world knowledge” actually come from the ability to perform longer reasoning chains.
3. The baselines and experimental details are insufficient to fully support the paper’s claims and analyses. Although the paper proposes an agent, WorldRetriever, it does not compare it against other agents or against strong long-reasoning models. In addition, the baselines appear to be relatively small models (around 7B) that are not specifically trained for reasoning. The paper also does not clearly describe how the LLM or VLM is prompted to answer the WorldQA questions.

Additional Questions:

1. The paper uses GPT-4 to estimate how many reasoning steps are required for each QA example. How trustworthy is this result? GPT-4 itself may have limitations in long-chain multimodal reasoning.
2. The paper states that it uses a ChatGPT model with 20B parameters. Does this refer to the gpt-oss-20b model?

**Audience:**

Yes

**Audience Explanation:**

I can see that the motivation behind WorldQA could be valuable for evaluating multimodal model performance, especially for models that emphasize long-chain reasoning. Researchers and practitioners working on multimodal models, particularly those focused on multimodal reasoning, may find this dataset helpful. As a result, I believe that at least part of the TMLR audience would be interested in the findings of this paper.

**Claims And Evidence:**

No

**Claims Explanation:**

As discussed above, I find the motivation for the dataset convincing, and I agree that such a benchmark is needed. However, the “world knowledge” or “experiential knowledge” described and illustrated in the paper appears somewhat naive. This makes me question whether world knowledge itself is truly the key factor behind the improvement, or whether the gains instead come mainly from longer chains of reasoning that process this so-called expert knowledge.

On the other hand, the authors list WorldRetriever as another main contribution, but they do not compare it against other agents or against agents of a similar scale. As a result, the baselines seem insufficient for properly evaluating the effectiveness of the proposed WorldRetriever. More broadly, it does not seem entirely fair to compare baseline models that are neither fine-tuned nor specifically designed for long-chain reasoning against an agent that is explicitly prompted and constructed to perform long-chain reasoning.

**Requested Changes:**

1. I would like the authors to improve the organization and presentation of the paper, including the issues mentioned in the summary, as this would make the work easier to follow and understand.

2. I would like the authors to better justify the importance and effectiveness of “world knowledge” or “expert knowledge” in WorldQA. The current illustrations and examples are not fully convincing to me, as they seem to involve fairly naive or basic knowledge that strong baseline models may already possess, even without querying the internet or consulting an external expert source.

3. I would like the authors to investigate whether the observed performance gains attributed to “world knowledge” may instead come from longer chains of reasoning. Clarifying this point would help better establish what WorldQA is actually measuring.

4. I would like the authors to include another baseline agent with a similar purpose for comparison, in order to better demonstrate the effectiveness of WorldRetriever. If such a comparison is not feasible, the paper should more clearly explain why it is unnecessary.

Among these points, 2 and 3 are the most important for validating the motivation and claims of WorldQA, while 4 is important for evaluating WorldRetriever. Point 1 is comparatively less critical, but I would still recommend revising the paper structure and presentation.

---

### Review · Reviewer_n9QK · 2026-03-30

**Summary Of Contributions:**

This paper introduces a WorldQA benchmark that evaluates complex reasoning and video understanding. It requires an average of 4.45 reasoning steps per question, with visual/audio data in the identified five types of world knowledge. To tackle this complexity, WorldRetriever is proposed, an agent that uses a LLM to synthesize information of specialized visual, auditory, and external knowledge retrievals but detailed elaboration is missing. While it currently outperforms state of the art proprietary models, it still only achieves approximately 70% of human-level performance, highlighting significant remaining challenges in model consistency and the effective processing of multi video frames.

**Additional Comments:**

To summarize, this paper does not fully describe the proposed method (especially about WorldRetriever), annotation quality. In addition, the all tested models are outdated (considering the fast evolution of the field raises the concern on their findings) and do not provid solutions to address the identified limitations of the current state of the literature. Revision on these matters are highly recommended.

**Audience:**

Yes

**Audience Explanation:**

The paper identifies a significant limitation in current LMMs where performance often diminishes as the number of video frames increases, contrasting sharply with human performance, which scales positively with additional visual context.

Evaluations reveal an inconsistency in prominent instruction-tuned models, where accuracy in multiple-choice tasks drops significantly because models fail to consistently select the same answer when option sequences are permuted.

For researchers who focus on common sense and cognition, this paper can provide a robust five-category framework for world knowledge, covering societal norms, self-motivation, tool use, social interactions, and multimodal associations to move benchmarks beyond simple object perception.

**Broader Impact Concerns:**

Please refer to the Requested changes section.

**Claims And Evidence:**

Yes

**Claims Explanation:**

The paper identifies a significant performance gap between current video/image-based LMMs and humans. By utilizing a "frame contrast" analysis, the authors demonstrate that while human accuracy improves with more visual data, current models often show diminished performance when processing an increased number of frames

The authors provide empirical evidence for the necessity of multimodal integration by performing ablation studies. By systematically removing audio and visual inputs, they demonstrate that the complex reasoning required for WorldQA cannot be solved through "language-only" shortcuts or by analyzing the natural language question in isolation.

The paper argues that WorldQA offers a more challenging benchmark than existing datasets by using "reasoning steps" as a core metric. To support this claim, the authors employed GPT-4 to map the logical hops required for each question, identifying an average of 4.45 steps, higher than the sub-two-step average found in prior datasets.

**Requested Changes:**

(major)
More elaboration is required regarding the design of the prompts in Table 2 and Table 3, as the current descriptions lack sufficient detail for reproducibility.

The criteria for partitioning reasoning steps remain ambiguous, and there is no evidence that the generated answers or reasoning steps were validated to filter out potential inaccuracies or hallucinations.

The main text lacks a comprehensive explanation of WorldRetriever. While Figure 3 illustrates the mechanism, it is not explicitly discussed in a dedicated paragraph. Also, only 2 lines of descriptions are reported on page 7. This makesit difficult to assess the technical novelty of the method.

The evaluation primarily benchmarks models from 2022 and 2023, such as FrozenBiLM, Otter-Video, VideoChat, Qwen-VL and LLaVA, which are now considered outdated. Inclusion of more contemporary open-source models is necessary for a relevant performance assessment.

The observation that image-based models outperform video-based models has been established in previous literature (e.g., IV-Bench), rendering this specific finding less surprising in the current research context

While the study analyzes language model size, the comparison is restricted to 7B models and proprietary systems. Evaluation across more diverse sizes, such as 13B or 32B, would provide a more robust understanding of scaling effects.

The study utilizes LLM-as-a-Judge (specifically GPT-4) to assess open-ended questions, yet the consistency and reliability of this metric are not thoroughly addressed.

While the study identifies critical issues (such as low response consistency and a significant gap between LMMs and human-level performance) it does not propose concrete solutions to mitigate these problems.

Although audio-only performance is reported, the study lacks broader ablations or detailed analyses regarding the specific impact of auditory cues on the reasoning process.

The videos in the WorldQA dataset are sourced from specific platforms including Ego4D, Epic-Kitchens, VidOR, and YouTube, which introduces potential inherent biases from these original data collections.

(minor)
Citation format shall be re-evaulated.

---

### Review · Reviewer_xG4W · 2026-04-12

**Summary Of Contributions:**

In this paper, the author proposes a new benchmark, WorldQA, to evaluate whether the model can combine multimodal information, world knowledge, and long-chain reasoning to complete question answering in video understanding tasks.
In addition, the author proposes a retrieval enhancement framework called WorldRetriever, hoping to demonstrate that introducing video key information retrieval and external knowledge retrieval can improve the performance of the model on this task.

Strengths:
1. This paper proposes a definition of world knowledge and studies the understanding ability of LMM from perspectives that is closer to the human interactive world.
2. Authors use carefully designed validation methods for the constructed benchmark to ensure its more comprehensive validation capability.
3. The proposed WorldRetriever combines multimodal knowledge to provide insights for large models to understand the complex world.

Weaknesses:
1. The construction of the dataset and the design of the proposed WorldRetriever are not clearly expressed enough.
2. The validation of world knowledge understanding is not intuitive. Although the annotation of benchmarks is based on the defined world knowledge, the method comparison does not analyze the performance of different types of models in different fields of knowledge.
3. The writing of this paper needs improvement.

**Audience:**

Yes

**Audience Explanation:**

Readers who focus on multimodal learning, video understanding, LLM evaluation, and benchmark construction in TMLR will be interested in this paper.

**Broader Impact Concerns:**

There are two main points that need to be supplemented:

1. Video source and privacy/copyright issues. The data includes publicly available video sources, and the author should provide a clearer explanation of the data usage, distribution, and potential privacy risks.
2. External knowledge retrieval may introduce bias and errors. The retrieved knowledge may not be accurate and may also have cultural biases, which can affect the model output. It is recommended to explain this in the Broad Impact section.

**Claims And Evidence:**

Yes

**Claims Explanation:**

Overall, the insights of the paper are mostly supported by clear evidence, but there are still some that are not solid enough.

Supported claims:
1. The existing models still have shortcomings in world knowledge understanding: this is indicated by the evaluation results.
2. WorldRetriever helps multimodal reasoning: this is shown by the evaluation metrics.
3. WorldQA evaluates long-chain multimodal world understanding: the experiment and analysis demonstrate that WorldQA presents a more challenging benchmark to LLMs and LMMs compared to existing approaches.

Not so solid claims:
1. The verification of world knowledge + long-chain reasoning ability: it is only reflected in the ideas of data construction, but not in comparisons, evaluations, or metrics.

**Requested Changes:**

1. Further verify whether WorldQA has truly measured the abilities of world knowledge understanding, while providing model analyses of such abilities using their detailed performance on the proposed benchmark. Authors should consider comparing the performance of all the mentioned models in vairous knowledge domains.
2. Provide more rigorous illustration for the dataset annotation process. The annotation process and specific annotation methods of annotators have not been claimed, and it is not clear how to make annotators meet the QA requirements of World Knowledge Understanding and Long Reasoning Chain.
3. Provide more detailed QA validation. The implementation of reasoning hop and multiple-choice QA generation should be clarified, rather than just showing a series of prompts.
4. Provide the specific implementation of WorldRetriever. This paper does not include supplementary materials, the author should consider submitting the missing details in the supplementary materials.
5. The parameter count of proposed WorldRetriever is somehow questionable. The parameter count of 20B is only for ChatGPT and it seems to exclude LLaVA, Beats, and Whisper.
6. Refine the paper writing:
    - Correct the claim of *Appendix 3.1* in Section 1.
    - Unify the tags of sections, figures, and tables in the manuscript. Use abbreviations (*Sec. 1*, *Fig. 1*, *Tab. 1*) or full names (*Section 1*, *Figure 1*, *Table 1*) uniformly. It should be ensured that there is a space before the numbering, rather than some having some or not.
    - Use references with parentheses (`\citep{}`) instead of inline references (`\citet{}`) when the reference is not the subject of the sentence.

---

### Decision · Action_Editor_JPxT · 2026-05-24

**Recommendation:** Reject

**Additional Comments:**

The authors are encouraged to address the raised issues and consider resubmission in the future.

**Audience:**

Yes

**Audience Explanation:**

TMLR audiences who focus on multimodal learning, video understanding, and benchmark construction for large language and vision models would be interested in the findings of this paper.

**Claims And Evidence:**

No

**Claims Explanation:**

While the reviewers acknowledged the motivation behind creating a benchmark for complex reasoning in video understanding, the paper suffers from several critical flaws that remain unaddressed due to the authors' failure to submit a rebuttal.

For example, a few prominent ones are as follows:

- The fundamental premise of measuring "world knowledge" is flawed, as reviewers noted the knowledge required in the dataset examples appears to be basic commonsense rather than expert knowledge. The paper fails to prove whether performance gains stem from this knowledge or simply from the implementation of longer reasoning chains.

- The empirical validation relies on outdated baseline models from 2022 and 2023. Furthermore, comparing the explicitly prompted WorldRetriever agent against small baseline models that are not designed for long-chain reasoning provides an unfair and incomplete evaluation.

- Crucial implementation details for WorldRetriever are missing from the text.

- The methodology also lacks a rigorous illustration of the dataset annotation process and prompt designs.

- The paper relies heavily on GPT-4 to assess reasoning steps and judge answers without adequately addressing the consistency of the metric or providing human validation to prevent inaccuracies.

Due to these critical concerns, combined with the lack of any author rebuttal to clarify or rectify these issues, the paper is not ready for publication.

**Resubmission Of Major Revision:**

The authors may consider submitting a major revision at a later time.